# Direct Electrochemistry of Glucose Dehydrogenase-Functionalized Polymers on a Modified Glassy Carbon Electrode and Its Molecular Recognition of Glucose

**DOI:** 10.3390/ijms24076152

**Published:** 2023-03-24

**Authors:** Yang Sun, Weishi Xue, Jianfeng Zhao, Qianqian Bao, Kailiang Zhang, Yupeng Liu, Hua Li

**Affiliations:** 1School of Life Sciences, Henan University, Kaifeng 475004, China; 2Engineering Research Center for Applied Microbiology of Henan Province, Kaifeng 475004, China

**Keywords:** glucose dehydrogenase, functionalized multi-walled carbon nanotubes, glassy carbon electrode, direct electron transfer, electrochemical behavior

## Abstract

A glucose biosensor was layer-by-layer assembled on a modified glassy carbon electrode (GCE) from a nanocomposite of NAD(P)^+^-dependent glucose dehydrogenase, aminated polyethylene glycol (mPEG), carboxylic acid-functionalized multi-wall carbon nanotubes (fMWCNTs), and ionic liquid (IL) composite functional polymers. The electrochemical electrode was denoted as NF/IL/GDH/mPEG-fMWCNTs/GCE. The composite polymer membranes were characterized by cyclic voltammetry, ultraviolet-visible spectrophotometry, electrochemical impedance spectroscopy, scanning electron microscopy, and transmission electron microscopy. The cyclic voltammogram of the modified electrode had a pair of well-defined quasi-reversible redox peaks with a formal potential of −61 mV (vs. Ag/AgCl) at a scan rate of 0.05 V s^−1^. The heterogeneous electron transfer constant (k_s_) of GDH on the composite functional polymer-modified GCE was 6.5 s^−1^. The biosensor could sensitively recognize and detect glucose linearly from 0.8 to 100 µM with a detection limit down to 0.46 μM (S/N = 3) and a sensitivity of 29.1 nA μM^−1^. The apparent Michaelis–Menten constant ($K_{m}^{\mathrm{app}}$) of the modified electrode was 0.21 mM. The constructed electrochemical sensor was compared with the high-performance liquid chromatography method for the determination of glucose in commercially available glucose injections. The results demonstrated that the sensor was highly accurate and could be used for the rapid and quantitative determination of glucose concentration.

## 1. Introduction

For the detection of glucose, a number of analytical methods, including high-performance liquid chromatography (HPLC) [1], photoelectrochemical approaches [2], colorimetric methods [3], and electrochemical analysis techniques [4] have been developed to date. Electrochemical biosensors have been used in biological, medicinal, and industrial analysis because of its specificity, high efficiency, simplicity, and quickness. Particularly, like CRISPR-cas aptamer-based sensor [5,6], molecularly imprinted polymers (MIPs) technology and enzyme-based sensors [7,8] have attracted a lot of attention. CRISPR-cas aptamers are more stable than antibodies or proteins. CRISPR-cas aptamer-based sensors can be stored at room temperature without losing functional activity [5]. Molecularly imprinted polymers (MIPs) first combine template molecules and functional monomers with covalent or noncovalent bonds through self-assembly. Then, they complete polymerization in the presence of cross-linking agents and initiators. Finally, the template molecules are removed, thereby forming specific binding sites or cavities that are complementary in size and shape to the template molecule [8]. This method has the advantages of tunability and environmental stability over traditional enzyme-based sensors. However, the lack of conductivity and electrocatalytic activity limits its application in non-enzymatic sensors [7]. The enzyme-based sensor can be directly used in complex samples due to the high selectivity of the enzyme system, and it has high sensitivity of the electrochemical electrode, which is popular on the biosensor market.

To improve the assay’s accuracy and reliability, mediator-type glucose oxidase (GOD) sensors were developed and tested on electronic receptors in vitro [9]. Its enzymatic reaction, however, is highly dependent on the concentration of dissolved oxygen as an electron acceptor [10,11,12]. In contrast, glucose dehydrogenase (GDH) is considered the most promising enzyme to replace glucose oxidase because it does not use oxygen as the electron acceptor [13], thus avoiding the issue. It is not limited by dissolved oxygen because such detection can lead to errors.

NAD(P)^+^-dependent GDH (EC 1.1.1.47) is a member of the SDR-reducing protein family and is composed of identical subunits similar to those of other short-chain reductases/dehydrogenases [14]. Each enzyme has a subunit mass of 28.09 kDa and dual cofactor specificity (both NAD^+^ and NADP^+^) [15,16]. The requisite cofactors NAD^+^ and NADP^+^ have no toxicity as electron acceptors for glucose [17,18,19]. NAD(P)^+^-dependent GDH can catalyze the production of β-D-glucose to D-glucose-δ-lactone and NAD(P)^+^ as a cofactor to NAD(P)H [20]. NAD(P)^+^-dependent GDH has been shown to have higher activity compared to many other oxidoreductases that depend on other redox cofactors such as flavin adenine dinucleotide (FAD) [21,22], pyrroloquinoline quinone (PQQ) [23,24], and nicotinamide-dependent GDHs. NAD(P)^+^-dependent GDH does not permanently bind the cofactor within the enzyme but instead acts as a soluble co-substrate, with hydrogen acceptors and donors for intermediates in the reaction sequence.

Direct electron transfer between the enzyme and the naked electrode surface is typically difficult in the development of enzyme biosensors due to the large spatial separation between the enzyme prosthetic group and the electrode surface. However, incorporating the redox enzymes into a special conductive film on the electrode surface facilitates the electron transfer of the redox enzymes [25]. In some cases, a redox mediator is required to improve the efficiency of electron transfer between the enzyme and electrode. These mediators shuttle electrons from electrochemically active sites which are buried inside the otherwise electrically insulating protein shell and electrode [26].

The inherent insolubility and hydrophobicity of MWCNTs pose a major challenge to their solution-phase operation. The approach for overcoming this problem is the noncovalent chemical functionalization of MWCNTs with functional polymers. Carboxylated functionalized multi-walled carbon nanotubes (MWCNTs-COOH) are used as the support reagents [27], and the adsorption of organic molecules onto the functionalized MWCNT surface may occur based on electrostatic, hydrophobic, and electron donor–acceptor (π–π) interactions as well as due to hydrogen bonds [28]. The presence of oxygen-containing groups promotes the exfoliation of CNT bundles and enhances their solubility in polar media as well as their chemical affinity with ester-containing compounds such as polyesters. The –COOH groups on the nanotube surface are useful sites for further modifications. Grafting can be performed by generating amide and ester bonds and attaching molecules, including synthetic and natural polymers. The functionalization of MWCNTs with a variety of groups such as –OH, –COOH, and –NH_2_ provides attachment sites for the adsorption of organic molecules [29].

Polyethylene glycol amine (mPEG-NH_2_) is a monoactive PEG derivative that can be used to carboxylate peptides and facile conjugation with MWCNTs-COOH can proceed via the coupling reaction between –COOH and –NH_2_ [30]. mPEG modification improves solubility and stability and reduces the immunogenicity of proteins and peptides. The ionic liquid (IL) 1-butyl-3-methyl imidazolium tetrafluoroborate has been shown to have high thermal stability and high conductivity [31]. Noncovalent linking between GDH and fMWCNTs-mPEG in an IL membrane can be performed and may help to preserve the stability of GDH’s structure and function [32]. Nafion solutions have excellent ionic conductivity, and are often used in catalyst coating and carriers, has excellent film-forming ability, has a wide range of applications in proton conduction, and helps researchers to improve the stability of nanomaterial-modified electrodes [33]. Because of its unique structure, Nafion has a very thin catalytic layer, which reduces the resistance to the transport of substances and has less influence on the conductivity of the electrode. The Nafion solution forms a polymeric film after air-drying, which has selective permeability. The polymer film formed using the Nafion solution can effectively protect the electrode from damage, and its unique properties offer numerous advantages for the electrochemical sensing of NAD(P)H-dependent dehydrogenase substrates [34].

GDH was used as the recognition unit in this study, as was previously suggested for immobilization on an aminated polyethylene glycol (mPEG), on carboxylic acid-functionalized multi-walled carbon nanotubes (fMWCNTs), and on IL composite functional polymer-modified glassy carbon electrode (GCE). The amino groups in PEG as well as the carboxyl groups in fMWCNTs and IL have better synergistic effects and thus are more effective at adjusting the hydrophobicity, stability, conductivity, and biocompatibility of the composite functional polymer film. Furthermore, the electrochemical behaviors of GDH on the modified GCE were also studied. The composite functional polymer could preserve the conformational structure and catalytic activity of GDH, resulting in the production of biosensors with high sensitivity, stability, and selectivity for glucose recognition and detection.

## 2. Results and Discussion

### 2.1. Characteristics of the Nanomaterials

SEM and TEM were used to characterize the fMWCNTs and mPEG. The morphological data of fMWCNTs and fMWCNTs-mPEG are shown in Figure 1A–D. Figure 1A depicts fMWCNTs coiling together to form large aggregates. fMWCNTs have large specific surface areas and serve as substrates for the material’s electrochemical behavior on the electrode surface. Figure 1B shows that mPEG is covalently bound to the fMWCNT surface, with a high loading capacity of fMWCNTs-mPEG for GDH molecules. In Figure 1D, fMWCNTs mixed with mPEG dispersed better than fMWCNTs alone in Figure 1C. mPEG prevents carbon nanotube agglomeration. mPEG enhances and adjusts the film-forming properties and dispersion of fMWCNTs. To prevent fMWCNTs from reuniting, the hydrophilic mPEG is mixed with them. On the electrode surface, mPEG disperses hydrophobic fMWCNTs. This increases the area of contact between GDH and nanomaterials; mPEG also improves the strength and stability of electrode membranes.

### 2.2. Electrochemical Behavior of Different Modified GCEs

No redox peak was observed at the bare GCE, as shown in Figure 2 (curve a). It had a much stronger signal than that of curve d, and the best signal was obtained from NF/IL/GDH/mPEG-fMWCNTs/GCE (curve e). The addition of fMWCNTs increased the current signal at the working electrode significantly, but its response signal was only present during the oxidation peak. There was no peak on the return sweep peak (curve b). The addition of mPEG improved the electrochemical signal (curve c), and the material’s formal potential (E°′ = E_pa_/2 + E_pc_/2) without the addition of GDH was −124 mV. The formal potentials corresponding to NF/GDH/mPEG-fMWCNTs/GCE (curve d) and NF/IL/GDH/mPEG-fMWCNTs/GCE (curve e) were both −61 mV versus Ag/AgCl. The addition of IL had no effect on electrical potential, and it could be used as an electrolyte support to promote the direct electron transfer of proteins. The IL was modified on the electrode surface to immobilize enzymes in the nanomaterial microenvironment, thereby synergistically increasing the catalytic activity of the modified electrode with mPEG. The observation that the IL can significantly increase the direct electron transfer rate on the electrode surface supported this conclusion [35]. The anodic peak current to cathodic peak current ratio (I_pa_/I_pc_ ≈ 1) indicates that the electrochemical process of the modified GC electrode is quasi-reversible [36]. Furthermore, the NF/IL/GDH/mPEG-fMWCNTs/GCE response signal is stronger than that of other composites and can thus be used in subsequent electrochemistry studies. The amino group in mPEG and the carboxyl group in the fMWCNTs may aid in homogenously dispersing the fMWCNTs in the functional membrane and improving the electrochemical properties of the proposed electrode. Nafion offers a good microenvironment for the electrode, protects it, and increases its stability.

### 2.3. Effects of Scan Rates

As the sweep speed increased, the electrochemical parameters changed. The cyclic voltammetry plots are shown in Figure 3A at a sweep speed of 0.1–3 V s^−1^. The CV method was used to investigate the electrochemical behavior of the modified electrode. Peak currents of the anode and cathode increased in a proportional manner as scanning speed increased.

The linearity increased with scan rate from 10 to 3000 mV s^−1^, as shown in Figure 3B. Peak currents were used to adjust the scan rate, indicating that the redox reaction is a surface-controlled electrochemical process and that the redox signals are generated by the recognition unit GDH immobilized on the fMWCNT layer. The following were the regression equations: I_pa_ (μA) = −94.735v (V s^−1^) − 2.5549 R^2^ = 0.9993 and I_pc_ (μA) = 92.871 v (V s^−1^) + 2.5549 R^2^ = 0.9991. This implies that the redox reaction is a surface-controlled electrochemical process and that the redox signals are generated by GDH immobilized on the fMWCNTs layer. This result demonstrates that the kinetic reaction dominates the electrode process.
(1)$$I_{p}=\frac{n^{2}F^{2}A\Gamma v}{4\mathrm{RT}}$$

According to the slope of peak current (I_p_) versus scan rate υ, A is the surface area (0.07065 cm^2^) of the modified electrode. The average surface concentration (Γ) of the GDH electroactive center on the GCE surface was estimated to be 4.04 × 10^−10^ mol cm^−2^, as in Equation (1) [25], which was greater than the theoretical Γ value of 1.7 × 10^−10^ mol cm^−2^ which was based on PDB data of GDH (PDB ID: 1gco, approximately 10 nm in diameter). Dense arrays of GDH molecules were more readily formed on the hydrophilic surfaces of the cross-linked carriers mPEG and fMWCNT rather than on bare glass electrodes.

The peak voltage was also affected as the scan speed increased. When the scan speed was sufficiently large, the natural logarithm and the peak voltage were linear. The peak potential was in the range of 1.2–2 V s^−1^, and the linearity of these methods was investigated using E_pa_ = −0.1184 ln(V) − 0.1341 R^2^ = 0.996 and E_pc_ = 0.0913 ln(V) + 0.0739 R^2^ = 0.9931 (Figure 3C).

According to Laviron’s equation Equations (2) and (3) [37,38]:(2)$$E_{p}=E^{O^{\prime }}+\frac{\mathrm{RT}}{\alpha \mathrm{nF}}-\frac{\mathrm{RT}}{\alpha \mathrm{nF}}\mathrm{lnv}$$
(3)$${\mathrm{lnk}}_{s}=\alpha \ln\left(1-\alpha \right)+\left(1-\alpha \right)\ln\alpha -\ln\left(\frac{\mathrm{RT}}{\mathrm{nFv}}\right)-\alpha \left(1-\alpha \right)\frac{{\mathrm{nF}\Delta E}_{p}}{\mathrm{RT}}$$
where α (0.3 < α < 0.7) is the cathodic electron transfer coefficient, n is the number of electrons, T is the temperature (293 K here), R is the gas constant (8.314 J K^−1^mol^−1^), and F is the Faraday constant (96,485 C mol^−1^).

Thus, the cathodic electron transfer coefficient (α), the apparent heterogeneous electron transfer rate constant (k_s_) value, and the number of electrons (n) were evaluated to be 0.5, 6.5 s^−1^, and 2, respectively.

According to the analysis of the natural logarithm of the root sweep velocity as a function of peak current (Figure 3D), the worse linear relationship which is exhibited by the root sweep velocity as a function of peak current indicates that the electron transfer in this modified system is controlled by kinetics rather than diffusion [39].

### 2.4. Effect of pH on the Peak Potential

The electrochemical process of the modified electrode is affected by the pH of the PBS solution in the test environment. The electrochemistry behavior of the NF/GDH/IL/mPEG-fMWCNTs/GCE was studied using CV at various pH levels. Figure 4A depicts the electrode’s peak currents of in PBS at pH levels ranging from 4 to 9. As the pH value increases, the position of the measured oxidation and reduction peaks shifts toward negative voltages, as shown in Figure 4B. Peak current increased as the pH increased from 4 to 7 and then decreased as the pH was increased from 7 to 9. At pH 7, the maximum cathodic current was obtained. The cathodic and anodic peak current ratio was closer to 1. Later investigations determined that a pH of 7.0 was the optimum value. Moreover, the formal potential was linear with pH from 4 to 9 (Figure 4C). This slope was 57.9 mV pH^−1^, which is close to the Nernstian value of 59.2 mV pH^−1^ [25,40]. These findings indicate that electrons’ direct electrochemical behavior in this electrochemical system is reversible [36]. The sensor response was caused by the enzymatic reaction described below:(4)$$\beta -D-\mathrm{glucose}+2{\mathrm{NADP}}^{+}+2e^{-}\overset{\mathrm{GDH}}{\to }D-\mathrm{glucose}-\delta -\mathrm{lactone}+2\mathrm{NADPH}$$

Figure 1 depicts the electrochemical process of the NF/GDH/IL/mPEG-fMWCNTs/GCE. The nicotinamide ring of NADP^+^ accepts two hydrogen ions and two electrons. The reduced form of this carrier generated in this reaction is called NADPH. It can be oxidized electrochemically as electron transfer proceeds.

### 2.5. Detection of Glucose Concentration in Samples

Linear sweep voltammogram (LSV) is a common method for detecting substances. The LSVs of the NF/GDH/IL/mPEG-fMWCNT/GCE are shown in Figure 5A in the presence of different concentrations of glucose (0.8–100 μM). The gradual addition of glucose resulted in a decrease in GDH on the modified GCE, which result in a decrease in peak current as the reaction progressed [41].

The apparent Michaelis–Menten activity ($k_{m}^{\mathrm{app}}$) indicates the enzyme–substrate kinetics of the biosensor:(5)$$\frac{1}{I}=\frac{k_{m}^{\mathrm{app}}}{I_{\max}}\frac{1}{C}+\frac{1}{I_{\max}}$$
where I is the steady-state current, I_max_ is the maximum current under stationary substrate conditions, $k_{m}^{\mathrm{app}}$ denotes the apparent Michaelis constant, and C is the glucose concentration. The I_max_ and $k_{m}^{\mathrm{app}}$ values were obtained from extrapolation. Extrapolation of the plot is shown in Figure 5B.

This is because of the low $k_{m}^{\mathrm{app}}$ of 0.21 mM, which demonstrates strong substrate binding ability and high glucose affinity to the modified electrode during the detection process. The selected composite nanomaterials can shorten the bridge distance between GDH and GCE.
LOD = 3 S_0_/S(6)
where S_0_ and S denote the standard deviation measured under the blank solution and the sensitivity of the biosensor, respectively. The detection limit is calculated to be 0.46 μM using Equation (6). Table 1 shows the significant advantages of the proposed biosensor over GDH biosensors reported in recent studies.

### 2.6. Comparison of Sensor and HPLC Methods for the Detection of Glucose

Glucose concentrations of 10, 20, and 50 μM were used. Before detection, the glucose-containing injection was removed from the impurities by filtration through a 0.22 μm aqueous microporous membrane. HPLC and an electrochemical sensor were used to investigate the glucose injection, peak currents and linear voltammogram curves of diluted glucose injection, which were recorded at an electrochemical workstation.

Glucose (purity 99%) was prepared with a 5 mM dilute sulfuric acid solution of glucose mixed standard solution. By HPLC the linear range was 0.36–18 μg mL^−1^, and the linear equation was y = 1707.8C (μg mL^−1^) − 177.7.

The experimental results are shown in Table 2. The constructed electrochemical sensor shows good accuracy while avoiding the problems of long sample preparation time and expensive test equipment associated with HPLC.

### 2.7. Anti-Interference Performance and Selectivity of Modified Electrode

The anti-interference ability of the modified electrode was investigated using the amperometric current–time (I-t) method [39,41]. When the carbon source and byproduct in fermentation interference substances, such as sucrose, lactose, ethanol, propanol and lactic acid with a concentration of 200 μM were added to the detection solution, there was no obvious response (Figure 6). When glucose (10 μM and 100 μM) was added, a strong signal response occurred, indicating that the electrochemical sensor had good selectivity and anti-interference properties. This is the enzyme-based biosensor’s unique advantage, it has high enzyme selectivity, can directly determine glucose in complex samples, and has high electrochemical electrode sensitivity.

### 2.8. Characterization of Composite Functional Polymer

It is very critical to properly assemble the restricted enzyme elements during the preparation of enzyme electrodes without affecting their catalytic activity. The initial catalytic reaction rates were examined to investigate the effect of different materials on the catalytic activity of GDH. All nanomaterials reacted with GDH at room temperature for 20 min. The operation in question occurred as follows: a mixture of nanomaterials and GDH was added to PBS, and then NADP^+^ was added to initiate the reaction. We investigated the initial reaction rate of the product NADPH at the characteristic absorption peak at 340 nm [49]. Here, GDH bound NADP as a cofactor, and its reduced form could not reduce molecular oxygen, thus eliminating the interference of oxygen.

Only the addition of fMWCNTs affects the initial reaction rate of GDH, as shown in Figure 7A. The initial reaction rate of GDH can be significantly enhanced upon the addition of IL and mPEG. The initial reaction rate of GDH is fastest after mPEG/fMWCNTs/IL is mixed with GDH. IL can be used as a supporting electrolyte for promoting the direct electron transfer of the protein to the electrode [25]. IL and mPEG could improve fMWCNTs biocompatibility with GDH without changing GDH conformation.

The fluorescence spectra were recorded using an Infinite F Plex Fluorescent multi-mode enzyme marker. For full-wavelength scanning, the excitation wavelength was 270 nm, and the emission wavelength band was set at 280 to 400 nm. According to Figure 7B, mPEG, IL, and fMWCNTs could only lead to a minor red shift (less than 5 nm) [38] for the maximum fluorescence emission wavelength of GDH. As a result, there was no significant interaction between these materials and GDH. Composite nanomaterials provide a microenvironment for reaction. They do not react strongly with GDH, do not change the conformation of GDH, and have good biocompatibility.

EIS proved to be an effective method for probing the electrical properties of the interface between the modified electrodes and electrolyte [50]. The Nyquist plot is composed of two parts: a semicircular part and a linear part. The semicircular part corresponds to the electron transfer-limited case at higher frequencies, while the linear part at lower frequencies represents the electron diffusion-limited process on the electrode surface [51].

The EIS spectra are presented as Nyquist plots (Z′ versus Z″) on bare GCE, fMWCNTs/GCE, mPEG-fMWCNTs/GCE, mPEG-fMWCNTs/GDH/GCE, and mPEG-fMWCNTs/IL/GDH/GCE in 5 mM [Fe(CN)_6_]^3−/4−^ solution (Figure 8). Obviously, there were significant differences in the impedance spectra at different electrodes. The R_ct_ was estimated to be 86.56 Ω, 95.03 Ω, 101.30 Ω, 103.9 Ω, and 161.4 Ω. The R_ct_ significantly increased after the addition of nanomaterials. The addition of IL to the modified interface could be attributed to more negative charges on the surface. At low frequencies, the added electrons (a, b, c) diffuse freely on the surface of capillary electrophoresis without the addition of enzymes. The transfer of electrons to the surface of capillary electrophoresis is linear. The (d, e) electrons on the surface of GCE are controlled by dynamics rather than diffusion after the addition of GDH.

### 2.9. Stability of NF/IL/GDH/mPEG-fMWCNTs/GCE

The stability of the sensors was evaluated by monitoring the cyclic voltammetry peak currents of GDH after 100 cycles of continuous scanning (Figure 9A) and long-term storage (Figure 9B). After 100 cycles, the peak current (I_pa_ and I_pc_) remained at 98.05% and 98.80%, respectively, compared to the first cycle. There was no discernible decrease in the cyclic voltammetry response, demonstrating the acceptable durability of the biosensor in the buffer solution.

Long-term storage stability is another important factor affecting the commercial use of the sensor. The modified electrodes were stored in a desiccator at 4 °C. We then further studied peak currents over the course of 28 days by cyclic voltammetry. The peak current decreased because enzyme activity decreased during storage, and then the current density slowly decreased. The peak current lost approximately 18% of its original value, thus indicating that the modified GCE was stable in storage. Nafion polymer films effectively protected the electrode from damage.

## 3. Materials and Methods

### 3.1. Reagents and Materials

NAD(P)+-dependent GDH (EC 1.1.1.47, from *Bacillus subtilis*) was derived from directed evolution in our laboratory (Kaifeng, China) [49]. NaH_2_PO_4_, Na_2_HPO_4_, NADP^+^, sucrose, lactose, ethanol, propanol, lactic acid and β-D-(+)-glucose were purchased from Sigma (St. Louis, MO, USA). The mPEG (MW:2000), IL (1-butyl-3-methylimidazolium tetrafluoroborate), Nafion (NF) and functionalized multi-wall carbon nanotubes (fMWCNTs) were bought from Xian Ruisi Biological Technology, Co., Ltd. (Xi’an, China), Sichuan West Asia Chemicals Co., Ltd. (Chengdu, China), and Shenzhen Nanotech Port Co., Ltd. (Shenzhen, China), respectively. The glucose stock solution was kept for 24 h before use. All reagents were of analytical grade and used without further purification. All water used in this study was prepared to 18 MΩ via an ultrapure water machine.

### 3.2. Preparation of Functional Polymer-Modified GCE

Before coating, the GC electrode was mechanically polished twice with alumina (particle sizes 1.00, 0.30, and 0.05 μm) to the level of a mirror finish. After sonication for 10 min in 75% ethanol and double-distilled water, the GCE was dried for 30 min at 25 °C [39]. The electrode was then treated electrochemically in 0.2 M sulfuric acid with cycling between −0.6 and +0.4 V (vs. Ag/AgCl) at a sweep rate of 0.05 V s^−1^ for approximately 20 min.

### 3.3. Biosensor Construction and Measurement

The preparation process of the functional polymer-modified GCE was as follows (Figure 2): a mixture solution of mPEG (8 mg mL^−1^) and fMWCNTs (2 mg mL^−1^) (volume ratio = 1:8) was incubated for 4 h at room temperature. After 20 min of ultrasonic treatment, 2 µL of the mixture solution was dropped onto the surface of the prepared GCE and dried at room temperature for approximately 30 min. Then, 1 µL of IL was dropped onto the electrode. After the electrode was stored at 4 °C for 8 h, the NADP^+^ (50 mg mL^−1^) was mixed with GDH (4 mg mL^−1^). Then, 2 µL of the admixture was dropped onto the surface of the electrode and dried in the drying tower at room temperature for 2 h. Finally, 2 µL of 5%NF was dropped onto the electrode for protection.

### 3.4. Apparatus and Measurements

Electrochemical investigations were performed with a CHI660E system (Shanghai Chenhua Instrument Co., Ltd., Shanghai, China) in a conventional three-electrode system. An Ag/-AgCl-saturated KCl electrode, a Pt wire, and a 3-mm-diameter GCE served as the reference, counter, and working electrodes, respectively. The electrochemical measurements were performed in N_2_-saturated PBS (50 mM, pH 7.0) at 25 ± 1 °C. Fe(CN)_6_^3−/4−^ (5.0 mM) and KCl (0.1 M) were used for the electrochemical impedance spectroscopy (EIS) measurements at a bias potential of 0.20 V under the frequency range of 10^2^ to 10^6^ Hz. Scanning electron microscopy (SEM, JSM-7610F Plus, JEOL Co., Ltd., Tokyo, Japan) and transmission electron microscopy (TEM, JEM-2100, JEOL Co., Ltd., Tokyo, Japan) were used to collect the images. The catalytic activity of GDH was studied on ultraviolet-visible (UV-Vis) spectrophotometer (TU-1901, Purkinje General Instrument Co., Ltd., Beijing, China). The conformational effect of nanomaterials on GDH was studied using a fluorescent multi-mode enzyme marker (Infinite F Plex, Tecan Group Ltd., Männedorf, Switzerland).

The content of glucose was measured using an HPLC instrument Waters 1515 with a refractive index detector Waters 2414 (Waters Corporation, Milford, MA, USA). Glucose standard samples and samples containing glucose to be tested were filtered (0.22 μm filter) before use with an Aminex HPX-87H column (300 mm × 7.8 mm, 9 μm; Bio-Rad Chemical Division, Richmond, VA, USA). Sample peaks were analyzed using the Breeze software. The flowing phase was a 5 mM H_2_SO_4_ solution at a flow rate of 0.6 mL min^−1^ and the column was operated at 60 °C. The injection volume of the sample was 20 μL.

### 3.5. Nanomaterials and Glucose Dehydrogenation Enzyme Compatibility

Spectrophotometry is a low-cost and quick analytical method. NADH and NADPH have a characteristic absorption peak at 340 nm. Thus, GDH activity was measured by monitoring the production of NADPH [52,53]. Based on the electrode modification ratio, GDH reacts with nanomaterials at room temperature for 4 h to explore biocompatibility. The specific operation is as follows: 3300 μL of PBS (50 mM, pH 7), 400 μL of 1 M D-glucose solution, 100 μL of the GDH and nanomaterials mixed solution, and 60 μL of NADP^+^ were added in the cuvette. Then, we added NADP+ to initiate the reaction; the experiment was performed at room temperature (25 ± 1 °C); and the change in absorbance at 340 nm was measured using a TU-1901 UV-vis spectrophotometer. Wavelength changes and conformational effects of nanomaterials on GDH were studied via a fluorescent multi-mode enzyme marker.

## 4. Conclusions

In summary, we proposed a strategy for glucose detection based on Nafion, IL, mPEG, fMWCNT and GDH composite modified glassy carbon electrode. The composite polymer membranes were characterized by cyclic voltammetry, UV-vis spectrophotometry, fluorescence spectroscopy, electrochemical impedance spectroscopy, scanning electron microscopy and transmission electron microscopy. The heterogeneous electron transfer constant (k_s_) of GDH on the modified GCE is 6.5 s^−1^ in this study of electrochemical behaviour, which is a level that can detect glucose molecules. For glucose recognition, it has a good response range of 0.8 μM 100 μM and LOD = 0.21 μM. For the detection of commercially available glucose injections, we compared the electrochemical sensor with HPLC; the sensor demonstrated good accuracy and simplicity. Based on the characteristics of the glucose dehydrogenase sensor, such as its high sensitivity, strong anti-interference, and stable storage time, it provides a new concept for the application of the glucose dehydrogenase sensor in the future in the measurement of blood glucose, food safety, and fermentation engineering.

## Data Availability

Not applicable.

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
