# Peer review of "Direct Electrochemistry of Glucose Dehydrogenase-Functionalized Polymers on a Modified Glassy Carbon Electrode and Its Molecular Recognition of Glucose"

_ijms, 2023, doi:10.3390/ijms24076152_

Round 1

Reviewer 1 Report

In this paper, a novel glucose biosensor is synthesized by layer upon layer assembly of NF/GDH/IL/mPEG-fMWCNTs functional polymer to GCE. NAD(P)+ -dependent glucose dehydrogenase on glassy carbon electrode was studied. Direct electron transfer between GDH and GCE was realized.

Minor reversion is needed.

1. Ionic liquids are usually liquid at room temperature and are difficult to dry completely. How to ensure its stable existence on the electrode without falling off the electrode surface?

2. What is the concentration of Nafion solution used in the experiment?

3. Table 1 has to be improved by more literature LOD values.

4. Please add a set of control experiments to exclude the effect of NADP + peak on the electrode.

5. Please explain the reason why the current signal increases after adding GDH.

Reviewer 2 Report

The study demonstrated a glucose biosensor with high accuracy and potential in applied field to detect glucose quickly in quantitative manner and at low concentrations.

Few comments:

Provide LOD and Kd values in conclusions and abstract.

Introduction needs to broadened to cover other methods and specifically authors can cover some latest techniques of CRISPR-cas and aptamer-based sensing (10.1186/s13765-023-00771-9 and https://doi.org/10.1016/j.teac.2022.e00184).

Please elaborate caption of figure 1.

Scheme 1 is missing (not shown in the draft).

Scheme 2 could explained well with adding several details and by providing better resolution figure.

Please suggest future implications of such study on biological sensing of glucose.

Reviewer 3 Report

The authors presented direct electrochemistry of glucose dehydrogenase-functionalized polymers on a modified glassy carbon electrode and its molecular recognition of glucose. Although being interesting, I find that there are some major issues with the paper that require addressing prior to this being considered for publication in this journal. I have identified the main points for consideration below:

1.     This manuscript has some spelling typos, style errors and grammatical errors. Therefore, I advise the authors carefully check the whole manuscript and correct them.

2.     The novelty of this study should be clearly clarified in the manuscript.

3.     The selectivity of the proposed molecularly imprinted electrochemical sensor should be investigated. In addition, the selectivity coefficients for all interferences should be also determined.

4.     Imprinting coefficient is an important parameter for molecularly imprinted polymers. So, the imprinting coefficient of the MIP film should be added in the revised manuscript. 

5.     The applicability of the proposed sensor should be validated by real sample, and the results should be validated by the classic detection techniques.

6.     In the introduction section, some related references related to MIP-based sensors are recommended to be cited, such as Journal of Hazardous Materials 436 (2022) 129107; Materials Today Chemistry 26 (2022) 101043.

7.     The author should make a comparison on the sensing performance for glucose detection between the MIP- and NIP-based sensors.

8.     The sensing performance of this proposed MIP-based sensor should be compared with the previously reported ones.

Round 2

Reviewer 2 Report

Authors have improved the manuscript draft to acceptable level.

Reviewer 3 Report

The authors have addressed all comments, so I recommend it for publication in this journal.